# Telehealth versus face-to-face visits: A comprehensive outpatient perspective-based cohort study of patients with kidney disease

Lagu A. Androga[1], Rachel H. Amundson[1], LaTonya J. Hickson[1], Bjoerg Thorsteinsdottir[2], Vesna D. Garovic[1], Sandhya Manohar[1], Jason K. Viehman[3], Ziad Zoghby[1], Suzanne M. Norby[1], Andrea G. Kattah[1], Robert C. Albright, Jr[1]*

1 Nephrology & Hypertension Division, Mayo Clinic College of Medicine, Rochester, Minnesota, United States of America, 2 Community Internal Medicine, Mayo Clinic College of Medicine, Rochester, Minnesota, United States of America, 3 Quantitative Health Sciences, Mayo Clinic College of Medicine, Rochester, Minnesota, United States of America

☯ These authors contributed equally to this work.
* albright.robert@mayo.edu

**Data Availability Statement:** All relevant data are within the paper and its Supporting Information files.

## Abstract

### Background

Telenephrology has become an important health care delivery modality during the COVID-19 pandemic. However, little is known about patient perspectives on the quality of care provided via telenephrology compared to face-to-face visits. We aimed to use objective data to study patients' perspectives on outpatient nephrology care received via telenephrology (phone and video) versus face-to-face visits.

### Methods

We retrospectively studied adults who received care in the outpatient Nephrology & Hypertension division at Mayo Clinic, Rochester, from March to July 2020. We used a standardized survey methodology to evaluate patient satisfaction. The primary outcome was the percent of patients who responded with a score of *good (4)* or *very good (5)* on a 5-point Likert scale on survey questions that asked their perspectives on *access* to their nephrologist, relationship with *care provider*, their opinions on the *telenephrology technology*, and their *overall assessment* of the care received. Wilcoxon rank sum tests and chi-square tests were used as appropriate to compare telenephrology versus face-to-face visits.

### Results

3,486 of the patient encounters were face-to-face, 808 phone and 317 video visits. 443 patients responded to satisfaction surveys, and 21% of these had telenephrology encounters. Established patients made up 79.6% of telenephrology visits and 60.9% of face-to-face visits. There was no significant difference in patient perceived *access* to health care, satisfaction with their *care provider*, or *overall quality* of care between patients cared for via telenephrology versus face-to-face. Patient satisfaction was also equally high.

**Funding:** Mayo Clinic, Nephrology & Hypertension Research Committee - $5,000 funding for statistical personnel support for their time. The funders had no role in study design, data collection and analysis, decision to publish, or preparation of the manuscript.

**Competing interests:** The authors have declared that no competing interests exist.

## Conclusions

Patient satisfaction was equally high amongst those patients seen face-to-face or via telenephrology.

## Introduction

In early 2020, the novel Coronavirus (COVID-19) emergency resulted in the cancellation of more than 70% of face-to-face patient visits [1] as people sought to avoid infection by staying away from healthcare institutions, and many states instituted stay at home orders. Patients and providers had to find ways to connect while avoiding exposure to the virus. At the same time, the Centers for Medicare & Medicaid Services (CMS) temporarily broadened access to tele-health services, allowing all Medicare patients to request telehealth visits and providers to be reimbursed for more telehealth services [2]. As a result, in March–April 2020, consumer and provider adoption of telehealth services as a replacement for cancelled face-to-face visits increased 300-fold at the height of the pandemic when compared to 2019 [3].

Telehealth has become a valuable tool to leverage specialized medical care not just in areas with limited access to expert providers such as nephrologists, neurologists and intensivists but also during extraordinary circumstances like the COVID-19 pandemic which demands social distancing measures. There is, however, a dearth of objective data on patient perceptions of the quality of care provided via telehealth, and in particular, telenephrology, compared to patient perceptions regarding quality of care provided via face-to-face visits. The types of medical services and patients that would be appropriate for telenephrology remain unclear. There is limited data on the types of telenephrology health care delivery models (telehealth only versus face-to-face interactions supported by telehealth) that would best serve the needs of patients while reducing health care costs. There is also still little known regarding long term patient and provider perspectives on the use of telenephrology.

In this retrospective study, we review the experience of the Division of Nephrology & Hypertension at Mayo Clinic, MN, with the use of telenephrology for outpatient visits from March to July 2020. We compared patient satisfaction across telenephrology (phone or video) and face-to-face visits utilizing a standardized structured uniform survey methodology. We evaluated the patient characteristics, spectrum of kidney diseases with diagnosis codes and geographical locations of patients served via telenephrology.

## Methods

### Clinic model and setting

Mayo Clinic is a multispecialty academic primary care and quaternary referral center, with comprehensive kidney care provided for the entire local population as well as serving as an international destination medical practice across multiple specialties. The Mayo Clinic Rochester Division of Nephrology and Hypertension evaluates and manages acute consultative as well as longitudinal care of patients with all manner of kidney conditions.

This retrospective survey-based study was limited to patients seen in the outpatient clinic of the Division of Nephrology & Hypertension at Mayo Clinic, Rochester, MN. Kidney and/or pancreas transplant, inpatient and end-stage kidney disease patient visits were excluded from the study. The providers included 48 Nephrology physicians and 4 Nephrology nurse

practitioners (NP); the NPs saw patients in a specialized chronic kidney disease (CKD) clinic that serves a local population.

The study was approved by the Mayo Clinic Institutional Review Board and deemed to be minimal risk research. However, only patients who were at least 18 years of age and had consented through the Minnesota Research Authorization process that their medical data be used for research were included in the study. Patient demographic information that included age, gender, race, and home geographical location were obtained from Epic EHR software (Epic Systems, Verona, WI). We used the results of the Press Ganey Medical Practice Survey obtained through the Mayo Clinic Office of Patient Experience. Each survey result was linked to a patient medical record number and date of Service. Nephrology billing data was obtained and, using medical record number and date of service, a unique ID that allowed us to identify the primary billing diagnosis from the revenue data was created. Individual patient responses to survey data were not shared with the investigation team to protect the confidentiality of the survey respondents, and insights are only shared at an aggregate level. Primary diagnoses were prospectively categorized into seven distinct groups, namely BP/Hypertension, CKD stage 1–3, CKD stage 4–5, cystic kidney disease, nephrolithiasis, kidney parenchymal disease, and other. See S2 Table in S1 File for comprehensive list of primary visit diagnoses. Patients could have multiple diagnoses and there was overlap in patients' diagnoses but for the purposes of our research, the primary diagnoses noted in the billing code supersede other kidney diagnoses for that particular visit.

## Patient satisfaction data

The Mayo Clinic Office of Patient Experience (OPE) Team uses Press Ganey (South Bend, IN), a third-party vendor, to create patient satisfaction surveys, randomly select patients to receive the surveys, and to collect and analyze patient satisfaction data. The surveys have been tested in 3,361 patients from five organizations over a five month period [4] and examined for validity, reliability and readability [5–8]. Surveys were emailed to patients seen either by phone, video or face-to-face. Each provider at Mayo Clinic, Rochester, MN is limited to having 30 of their patients surveyed per month though they may see more than 30 patients per month. Each patient receives only one survey for a visit to Mayo Clinic, irrespective of the number of medical specialties they visit. Patients had up to six weeks to respond to the surveys. The survey questions were grouped into 4 domains of patient experience, namely (1) access (2 questions), (2) care provider (5 questions), (3) telemedicine technology (3 questions), and (4) overall assessment (2 questions). S3 Table in S1 File lists the question content summaries and the patient experience domains assessed. Patients answered individual questions with a categorical 5-point Likert scale ranked from *very poor (1)*, *poor (2)*, *fair (3)*, *good (4)* and *very good (5)*. The Mayo Clinic OPE team provided aggregated patient survey reports for patients seen in the Rochester Division of Nephrology & Hypertension outpatient clinic from March 1st–July 31st 2020. The aggregated survey data presented the frequency of each of the 5-point Likert scale responses for each question for each patient who responded to the survey for a particular visit for outpatient Nephrology care.

## Outcomes

The pre-determined primary outcome was the percentage of patients who gave a score of *good (4) or very good (5)* on a Likert scale; a score of *good (4)* or *very good (5)* represent the two highest levels of patient satisfaction for each survey question. The pre-determined secondary outcome was the percentage of patients who scored *very good (5)*, representing the highest level of patient satisfaction possible.

## Statistical analyses

Continuous variables are presented as median (IQR), and categorical variables are described as number (percent). Phone, video, telenephrology (phone and video), and face-to-face visits are detailed separately, but were analyzed only as telenephrology versus face-to-face. This combination of phone and video was validated by examining phone versus video responses and finding no significant differences. Wilcoxon rank sum and chi-square tests were used, as appropriate. Survey results were presented as number (%) that answered good (4) or very good (5) but were analyzed on an ordinal (1–5) scale using the Wilcoxon rank sum test. Analyses were done using SAS v9.4 (SAS Institute Inc., Cary, NC).

## Results

### Participant characteristics

There were 10,079 outpatient nephrology clinic encounters at Mayo Clinic, Rochester MN between March 1$^{st}$ and July 31$^{st}$, 2020. After applying the exclusion criteria, 4,611 patient encounters remained to be considered for the retrospective study (Fig 1). In 2020, 54.8% of the patients seen in the outpatient setting used Medicare/ Medicaid insurance.

Out of those included in the study, 3,486 (75.6%) of the patient were conducted face-to-face, 317 (6.9%) via video and 808 (17.5%) by phone. Established patients were defined as patients who had had at least one prior visit within the last three years with the nephrology department. Established patients comprised 941 (83.6%) of the telenephrology visits compared to 2104 (60.4%) of the face-to-face visits. There was a statistically significant difference between the telenephrology and face-to-face groups with respect to the type of patient encounter and age; the telenephrology patients were more likely to be established patients (p-value <0.001) and younger (median 65.5 years (IQR 51.8, 75.3) versus 66.7 (IQR 53.8, 75.3) (Table 1). There was no significant difference in gender or race between the telenephrology and face-to-face cohorts. The majority of the patients in the study were white.

### Survey results

Out of the 4,611 encounters, there were 3455 unique patients in the study cohort. 443 patients responded to the Mayo Clinic Office of Patient Experience surveys. The survey response rate for division for the period October 2019 –September 2020 was 30.2% (S5 Table in S1 File). 350 (79%) of the 443 patients were seen via face-to-face visits compared to 93 (21.0%) via telenephrology visits. Like in the general study cohort, a higher proportion (79.6%) of telenephrology visits were by established patients, while only 60.9% of the face-to-face patients who responded to surveys were established patients. A high majority of the patients are white and there was a significant difference in race between the telenephrology and face-to-face patients who responded to surveys (p-value 0.007) (Table 2).

Using a five-point Likert scale ranging from 1 = very bad to 5 = very good, surveyed patients expressed their perspectives with regard to *access* to their nephrologist, how they related to their *care provider*, and when relevant–their opinions on the *telemedicine technology*–and their *overall assessment* of the care they received during the nephrology visit.

There was no significant difference in patient survey responses for the above when telehealth was compared to face-to-face visits. A large majority of patients selected a Likert score of at least 4 or 5 for all survey questions (Table 3).

At the deployment of the video visits modality, patient satisfaction with the telenephrology technology was closer to 80%, suffered an apparent decline after the initial month but rebounded within three months (Fig 2, Table 4). Of the 4611 outpatient nephrology

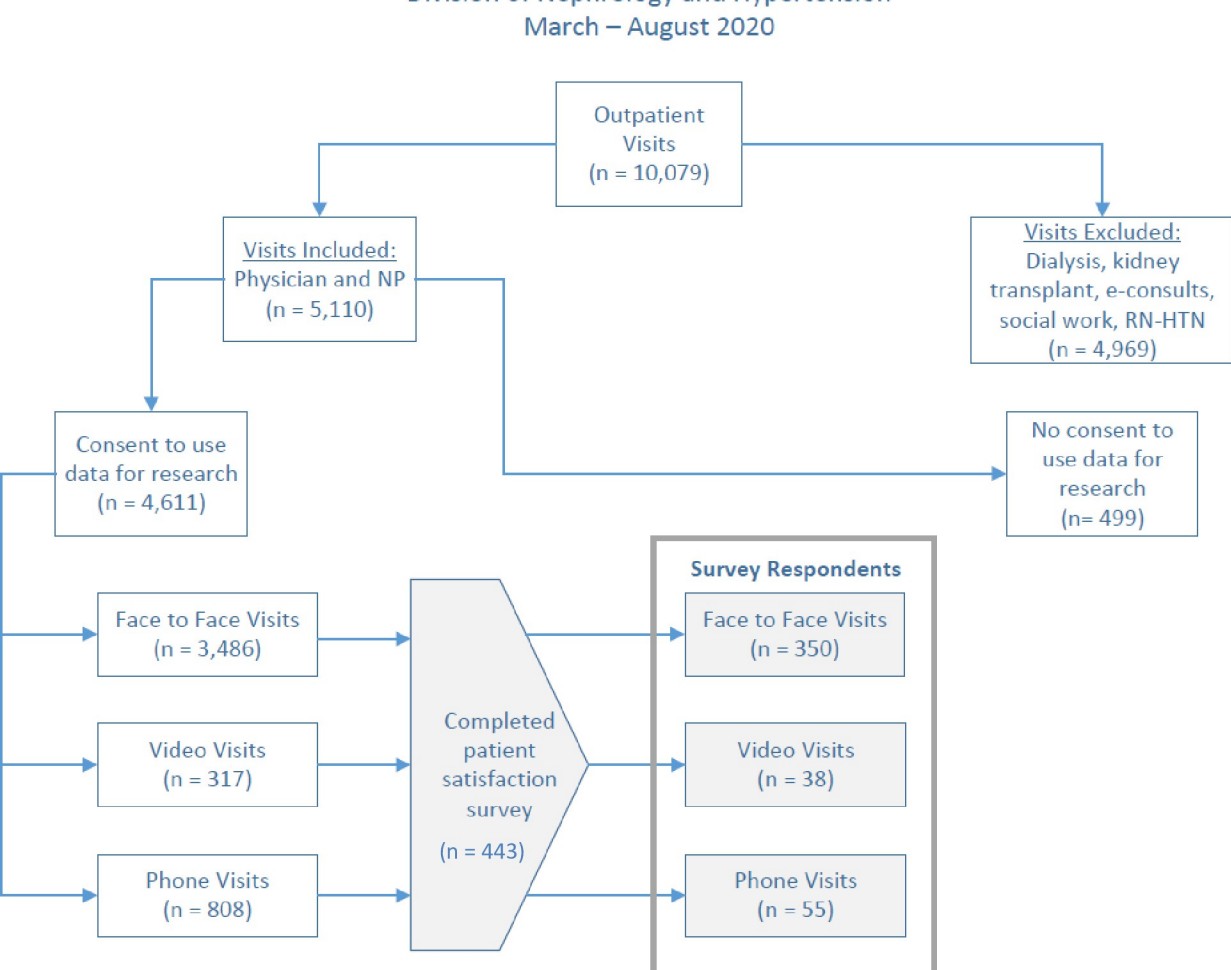

**Fig 1. Patient encounter selection criteria.** This schematic outlines the patient selection criteria.

encounters completed from March, 2020 –August 2020, there were 3455 patients, with 823 having more than one visit during the study period. 299 of the 823 were novel patients returning within the same initial encounter for short follow up visits.

## Travel distance saved by patients

19% (n = 214) of the patients seen via telehealth were from Rochester, MN while a majority were from outside Rochester but within 500 miles (n = 757, 67%). 13% (n = 148) were from more than 500 miles from Rochester, MN but within US boundaries, and only 1% (n = 6) were international patients. Fig 3

## Distribution of primary diagnoses

The patients were grouped into seven pre-determined primary diagnoses based on billing information. The primary diagnoses were CKD stage 1–3, CKD stage 4–5, BP/ Hypertension, kidney parenchymal disease, nephrolithiasis, cystic kidney disease, and other. CKD accounted for 36.5% (n = 411, 214 for CKD stage 1–3 and 197 for CKD stage 4–5) of the patients seen via

**Table 1. Study cohort patients' characteristics (March, 2020 –July, 2020).**

| Characteristics | Telenephrology | | | F2F | Total visits | P-Value |
|---|---|---|---|---|---|---|
| | Phone visits | Video visits | Combined | | | |
| **Number of visits, n (%)** | 808 (17.5%) | 317 (6.9%) | 1125 (24.4%) | 3486 (75.6%) | 4611 | |
| **n (%) of established patients** | 737 (91.2%) | 204 (64.4%) | 941 (83.6%) | 2104 (60.4%) | 3045 | <0.001 |
| **Age, Median (IQR), years** | 66.2 | 62.7 | 65.5 | 66.7 | | 0.015 |
| | (53.5, 75.0) | (47.4, 72.1) | (51.8, 73.7) | (53.8, 75.3) | | |
| **Gender n (%)** | | | | | | 0.880 |
| Female | 391 (17.8%) | 146 (6.7%) | 537 (24.5%) | 1655 (75.5%) | 2192 | |
| Male | 417 (17.2%) | 171 (7.1%) | 588 (24.3%) | 1831 (75.7%) | 2419 | |
| **Race, n (%)** | | | | | | 0.510 |
| White | 740 (17.5%) | 294 (7.0%) | 1034 (24.5%) | 3182 (75.5%) | 4216 | |
| Other | 68 (17.2%) | 23 (5.8%) | 91 (23.0%) | 304 (77.0%) | 395 | |

*p-values reflect Telehealth vs F2F groups, and are from Wilcoxon rank sum or chi-square tests as appropriate. F2F = face to face. Telenephrology (Combined) includes both phone and video visits.

telehealth, and the majority were seen via phone visits (Fig 4). Among the survey respondents, CKD accounted for 37% of patients seen via telehealth (n = 35, 24 for CKD stage 1–3 and 11 for CKD stage 4–5) (S2 Fig in S1 File).

## Discussion

This novel study with objective patient-based findings highlights an initial attempt to focus on the value equation of telenephrology. These authors are unaware of any published study set with specific patient satisfaction metrics in a comparison of telenephrology versus face-to-face Nephrology care. This retrospective survey-based cohort study demonstrates no observable significant difference in patient perceived access to health care, satisfaction with their care provider, or overall quality of care between telenephrology and face-to-face visits. Patient overall satisfaction was also equally high in telenephrology as compared to face-to-face visits. We also

**Table 2. Surveyed patients' characteristics (March–July 2020).**

| Characteristics | Survey respondents | | | | |
|---|---|---|---|---|---|
| | Telenephrology | | | F2F | P-value |
| | Phone visits | Video visits | Combined | | |
| **Number of visits** | 55 (12.4%) | 38 (8.6%) | 93 (21.0%) | 350 (79.0%) | |
| **n = 443** | | | | | |
| **n (%) of established patients** | 48 (87.3%) | 26 (68.4%) | 74 (79.6%) | 213 (60.9%) | <0.001 |
| **Age, Median (IQR), years** | 70.6 | 67.3 | 69.1 | 69.7 | 0.397 |
| | (61.2, 77.6) | (56.2, 72.1) | (58.1, 75.7) | (61.3, 77.1) | |
| **Gender, n (%)** | | | | | 0.100 |
| Female | 30 (16.1%) | 16 (8.6%) | 46 (24.7%) | 140 (75.3%) | |
| Male | 25 (9.7%) | 22 (8.6%) | 47 (18.3%) | 210 (81.7%) | |
| **Race, n (%)** | | | | | 0.007 |
| White | 51 (12.0%) | 34 (8.0%) | 85 (20.0%) | 341 (80.0%) | |
| Other | 4 (23.5%) | 4 (23.5%) | 8 (47.0%) | 9 (53.0%) | |

*p-values reflect Telehealth vs F2F groups, and are from Wilcoxon rank sum or chi-square tests as appropriate. F2F = face to face. Telenephrology (Combined) includes both phone and video visits.

Table 3. Frequency of survey top box scores: Likert score of 4 (good) or 5 (very good).

| | | Frequency of Top Box = 4 or 5 responses | | | | |
| --- | --- | --- | --- | --- | --- | --- |
| | | Telenephrology | | | F2F | |
| | | Phone | Video | Combined | | |
| | | n (%) | n (%) | n (%) | n (%) | P-value |
| **ACCESS** | Ease of scheduling your appointment | 49 | 38 | 87 | 327 | 0.056 |
| | | (90.7%) | (100%) | (94.6%) | (95.1%) | |
| | Ease of contacting us (e.g. email, phone, web portal) | 53 | 37 | 90 | 324 | 0.523 |
| | | (98.1%) | (97.4%) | (97.8%) | (94.7%) | |
| **CARE PROVIDER** | Concern the care provider showed for your questions or worries | 53 | 36 | 89 | 342 | 0.692 |
| | | (98.1%) | (94.7%) | (96.7%) | (98.6%) | |
| | Explanations the care provider gave you about your problem or condition | 54 | 35 | 89 | 341 | 0.637 |
| | | (100.0%) | (92.1%) | (96.7%) | (98.0%) | |
| | Care provider's efforts to include you in decisions about your care | 53 | 34 | 87 | 341 | 0.839 |
| | | (98.1%) | (89.5%) | (94.6%) | (98.3%) | |
| | Care provider's discussion of any proposed treatment (options, risks, benefits, etc.) | 53 | 34 | 87 | 342 | 0.400 |
| | | (100.0%) | (89.5%) | (95.6%) | (99.1%) | |
| | Likelihood of your recommending this care provider to others | 53 | 35 | 88 | 339 | 0.977 |
| | | (98.1%) | (92.1%) | (95.7%) | (96.9%) | |
| **TELEMEDICINE TECHNOLOGY** | Ease of talking with the care provider over the video or audio connection | 52 | 36 | 88 | | |
| | | (96.3%) | (94.7%) | (95.7%) | | |
| | How well the audio connection worked during your visit, whether by phone or video | 53 | 36 | 89 | | |
| | | (98.1%) | (94.7%) | (96.7%) | | |
| | If you had a video visit, how well the video connection worked | | 37 | 37 | | |
| | | | (97.4%) | (97.4%) | | |
| **OVERALL ASSESSMENT** | How well the staff worked together to care for you | 53 | 36 | 89 | 337 | 0.998 |
| | | (98.1%) | (94.7%) | (96.7%) | (98.3%) | |
| | Likelihood of your recommending our practice to others | 54 | 37 | 91 | 340 | 0.674 |
| | | (100.0%) | (97.4%) | (98.9%) | (98.3%) | |

*p-values reflect continuous scales and are compared between telehealth and F2F visits. A Wilcoxon Rank Sum test was used. F2F = face to face; Telenephrology (Combined) = Phone + Video

did not see any significant difference in patient perspectives when we compared patients evaluated via phone versus video visits.

Of note, over 87% of the phone compared to 68% of the video visits in the surveyed cohort were with established patients; although there was no observed statistically significant difference, satisfaction among the phone patients was also generally higher compared with video visit patients. The reason for the higher satisfaction among the phone patients deserves further study. This may be due to already established patients having favorable opinions about this medical center, its providers and perhaps through previous video and phone experiences. This finding is consistent with multiple research articles in which patient-physician familiarity is found to be associated with higher patient satisfaction [9–11].

Familiarity with and the ease of use associated with technology affected patient satisfaction. Satisfaction with video connection was initially high when the video program was rolled out. However, this dropped the following two months and improved back to high levels following an institution-wide process flow change. This institutional process change involved the video call setup being switched from an institutional centralized system to each department managing their own video setup process.

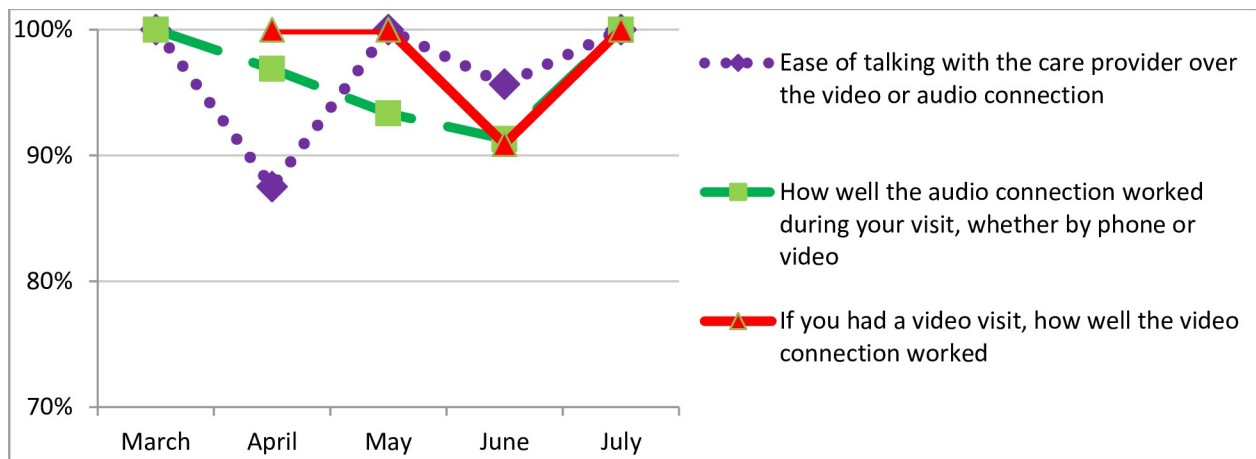

**Fig 2. Graph showing the frequency of trop box score of 4 or 5 by month for telenephrology responses.** Graphical illustration of satisfaction of audio, video and overall ease of communication with providers by survey month.

Additionally, improved satisfaction with the video technology may suggest the occurrence of a learning curve for the providers, through the 317 video visits over the span of five months. We speculate that patients' satisfaction with video visit technology also improved as they encountered more exposure to the virtual technology. We saw that 23% (n = 823) of our patients had more than 1 visit with the division during the 5-month period of the study. Hypothetically, patients may have become adept at using video technology, since they likely had video visits with other providers in other departments, institutions or even with family members, as tele-video visits became synonymous with social distancing.

There still remains a dearth of data on what type of diagnoses and types of patients are best served by telehealth. Our retrospective study shows CKD 1–5 made up 36.5% (n = 411), contributing the most to the number of telenephrology visits. 48% (n = 590) of the telehealth visits were also by patients who live over 100 miles from our medical center and 32% (n = 363) were from within 100 miles while only 19% (n = 214) were from Rochester MN. Our findings show that most of the telehealth patients live further away from Mayo Clinic and are generally more satisfied if they do not have to travel long distances to visit with their physicians.

This study has various limitations. Although overall satisfaction with telenephrology was high (>96%) and not statistically significantly different from face-to-face visits, the overall patient satisfaction survey response rate was quite low, introducing potential self-selection bias. However, these patient experience survey results are similar to the VA telehealth satisfaction results, where patient satisfaction among veterans was found to be equally high, with 92%

**Table 4. How the frequency of top box score of 4 or 5 varied by month for telenephrology survey responses.**

| Month | Ease of talking with the care provider over the video or audio connection | | | How well the audio connection worked during your visit, whether by phone or video | | | If you had a video visit, how well the video connection worked | | | Phone | Video |
|---|---|---|---|---|---|---|---|---|---|---|---|
| | Top box = 4 or 5 count | Total | % | Top box = 4 or 5 count | Total | % | Top box = 4 or 5 count | Total | % | n = 808 | n = 317 |
| March | 10 | 10 | 100% | 10 | 10 | 100% | | | | 124 | 2 |
| April | 28 | 32 | 88% | 31 | 32 | 97% | 9 | 9 | 100% | 331 | 62 |
| May | 15 | 15 | 100% | 14 | 15 | 93% | 7 | 7 | 100% | 112 | 73 |
| June | 22 | 23 | 96% | 21 | 23 | 91% | 10 | 11 | 91% | 141 | 92 |
| July | 13 | 13 | 100% | 13 | 13 | 100% | 11 | 11 | 100% | 100 | 88 |

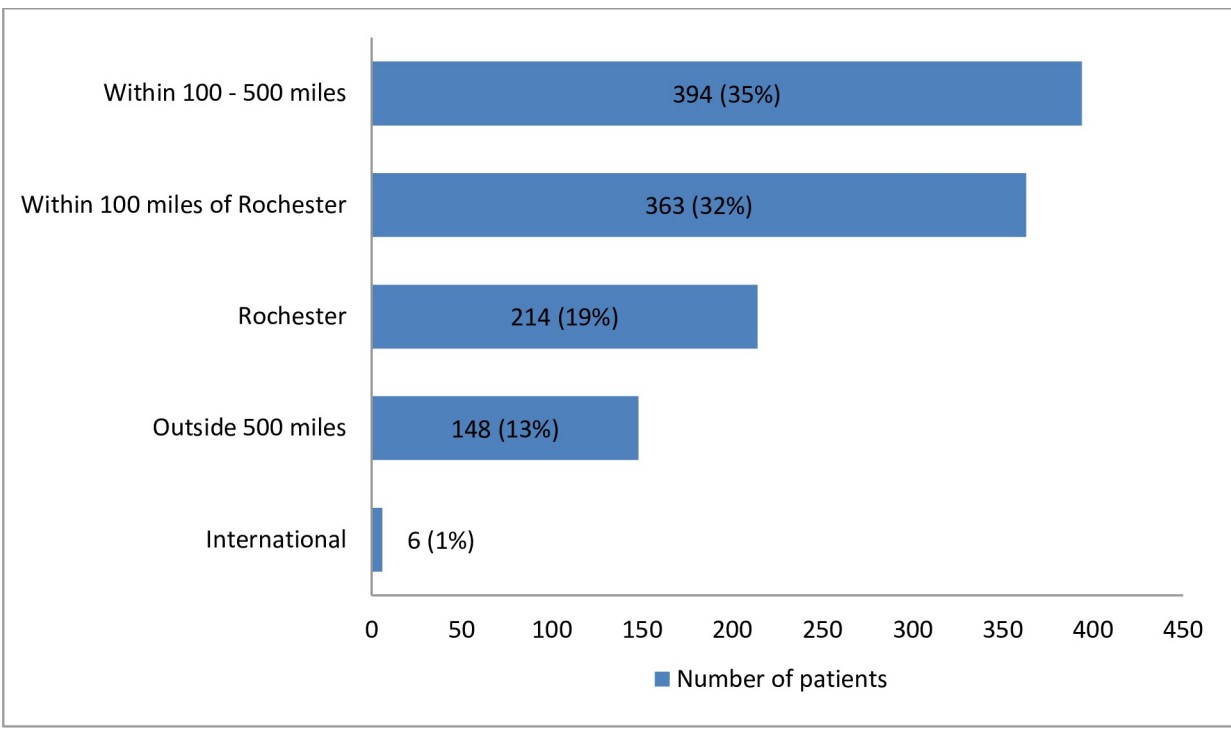

**Fig 3. Number and percent of telenephrology visits by geography.** The figure graphically represents the plurality of locations served by telenephrology.

for clinical video telehealth [12]. Since the Mayo Clinic patient experience team sends only one survey per patient and patients are more likely to have visits with more than one department, patients' experiences with other divisions likely contributed to their perceptions of telehealth. Our study therefore includes patients who were attended to in the Nephrology & Hypertension division but may have been seen by other medical departments too. To minimize the effect of recall bias, survey participants had up to 6 weeks to respond to the surveys. The risk of misclassification bias exists since we grouped the phone and video visit patients together, although the majority of the telenephrology visits were phone visits and also most of the phone visits were with established patients. We did compare the patient responses for phone vs video visits and saw no statistically significant difference between the two groups. This may suggest that combining phone and video visits was appropriate, but still deserves further research in the future. We also did not have the characteristics of survey non-respondents to compare how they may differ from respondents.

Survey-studied patient satisfaction is but one of the major ways of interpreting the overall value equation of patient outcomes. This study was limited in its ability to determine cost of care metrics and overall effectiveness of the care delivered. This latter issue is difficult to measure, but applications of guideline-driven medication prescription and care outcomes such as accepted metrics of diabetes, blood pressure and glomerular filtration rate (GFR) decline deserve tracking once more robust datasets are accumulated.

Our study period was also marked by the COVID-19 pandemic which on one hand spurred the widespread increase in the use of telehealth as a modality of health care delivery, but it may also suggest that patients were more accepting of telehealth services at this time opposed to when there are no concerns associated with risks posed by a pandemic. Potentially, patients' willingness to participate in telenephrology rather than face-to-face visits may drop once

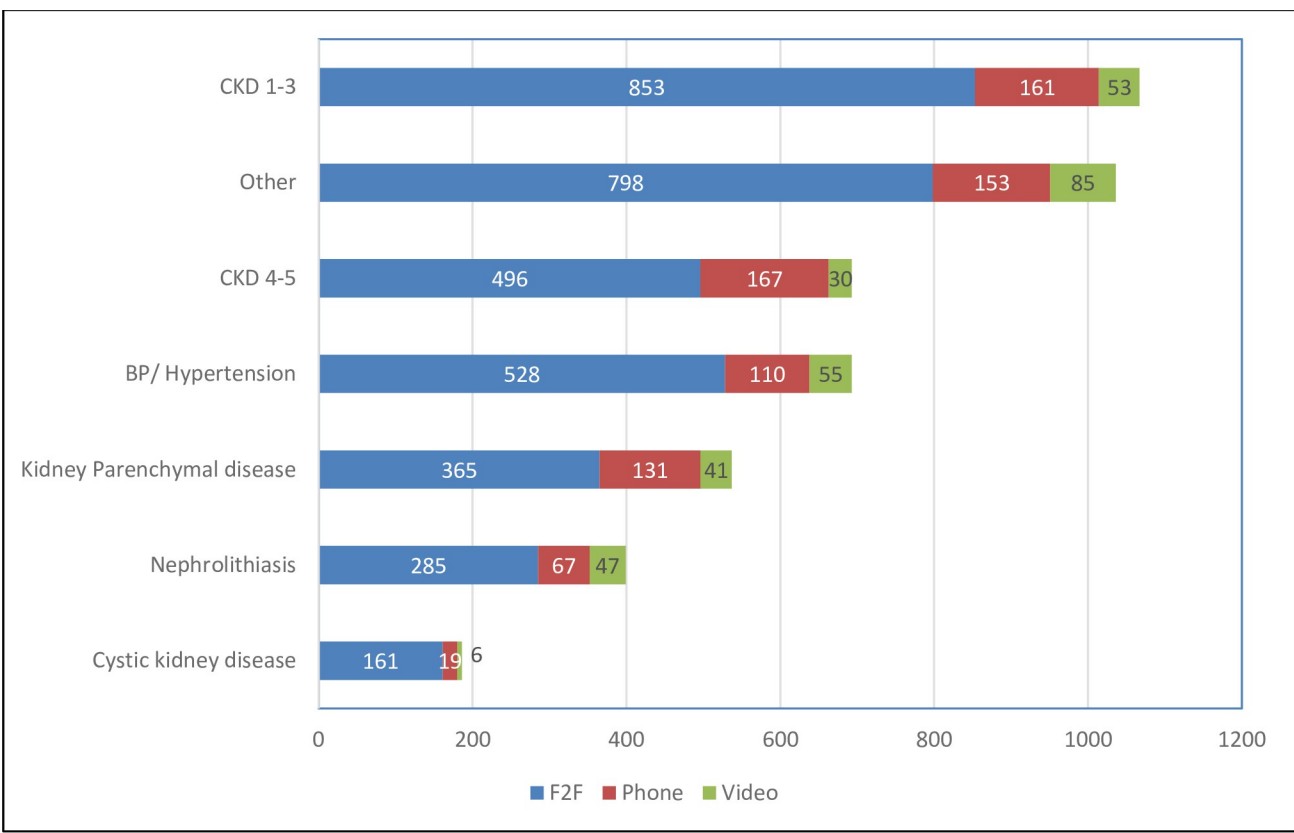

F2F = Face to Face; Telenephrology = Phone and Video

**Fig 4. Patient distribution by visit type and diagnoses groups (all patients, face to face and telenephrology).** The majority of patients in the study were best characterized as CKD, however, the entire spectrum of clinical visits was supported by telenephrology.

people no longer need to social distance. Whether acceptance of telenephrology for patients who live over 100 miles away and have limited access to nephrology care due to distance and travel costs will remain attractive deserves study as well in a post-pandemic era. The risk of acquiescence further complicates the results of our analyses. We saw many patients scoring *good (4) or very good (5)* for most questions and this may suggest the phenomenon whereby the patients agreed with all the questions or measures in the survey. Even though we used a survey that was designed and tested to reduce such bias, the risk and its inherent effect on interpreting the results persist.

The data presented highlight a difference in patient satisfaction over the time period of the study. The apparent decrease in time-based patient ratings with an initial high relative acceptance followed by a brief few weeks decline and ultimately a surge in high acceptance and satisfaction scores deserves further study and discussion. This was unlikely to have altered the overall outcomes based on the numbers of responses over these time frames. A few possible explanations may be entertained and may serve to highlight potentially promising avenues for future research efforts into various virtual practice processes. The initial time frame of telenephrology and other service lines at this institution were characterized by a central "rooming" process. This process was ultimately transitioned to the local service areas.

Its provisional and single center nature notwithstanding, the overall conclusion of our study is that patient satisfaction was equally high amongst those patients seen face-to-face or

via telenephrology. Whether telenephrology may be better-suited for established patients who are already familiar with the health care organization and providers, or for patients with specific diagnoses–and how the overall outcomes of clinical performance compare–will deserve robust study.

This study provides objective positive evidence that fostering telenephrology visits adds arguably the most crucial initial step necessary for more complete implementation of virtual care models in Nephrology.

## Supporting information

**S1 File. S1 Table: Number of available patient surveys per diagnosis group.** Distribution of diagnostic categories and survey responses; **S2 Table: Diagnoses Grouping of patients seen in Outpatient nephrology March–July 2020.** Overall diagnoses by number of patients seen in the division of Nephrology March through July 2020; **S3 Table: Outpatient Nephrology Survey.** Patient- facing survey tool; **S4 Table: How the frequency of top box score of 5 varied by month for telenephrology survey responses.** Variability of top box score: 5 only, reported by month; **S1 Fig: Press Ganey Survey—Medical Practice Survey Response Rate for October 2019 –September 2020, Nephrology Division, Mayo Clinic, Rochester MN.** Graphical form of the variability of the top box scores (4 or 5) by month; **S5 Table: Press Ganey Survey— Medical Practice Survey Response Rate for October 2019 –September 2020, Nephrology Division, Mayo Clinic, Rochester MN.** Nephrology provider numbers in the division of Nephrology and the survey response rates; **S2 Fig: Survey respondents distribution by visit type and diagnoses groups (Face-to-face and telenephrology).** Respondents by diagnostic groups; **S6 Table: Survey Respondents by primary diagnosis group.** Distribution of respondents by face to face and telemedicine.
(PDF)

## Acknowledgments

We are grateful for the assistance from the division of Nephrology & Hypertension at Mayo Clinic, the Mayo Clinic Office of Patient Experience (OPE) Team, the integrated cultures of the Mayo Clinic Health System and the division of Community Internal Medicine at Mayo Clinic, Rochester, MN.

L.A.A and R.C.A were involved in the project from the beginning, data collection, data analyses, manuscript writing and editing. R.C.A was the supervising author. A.L.A, L.J.H, B.T, V.D.G, S.M.N and R.C.A were involved in the conception of and design the research project and also participated in editing of the manuscript. A.G.K, Z.Z and S.M.N acted as advisors and were actively involved in editing of the manuscript. R.H.A assisted in data collection and creation of the diagrams and tables. J.K.V was the statistician on the project and was also involved in writing the statistical methods section.

## Author Contributions

**Conceptualization:** Lagu A. Androga, LaTonya J. Hickson, Bjoerg Thorsteinsdottir, Vesna D. Garovic, Ziad Zoghby, Suzanne M. Norby, Andrea G. Kattah, Robert C. Albright, Jr.

**Data curation:** Lagu A. Androga, Rachel H. Amundson, Robert C. Albright, Jr.

**Formal analysis:** Lagu A. Androga, Jason K. Viehman, Robert C. Albright, Jr.

**Funding acquisition:** Lagu A. Androga, Vesna D. Garovic, Robert C. Albright, Jr.

**Investigation:** Lagu A. Androga, Robert C. Albright, Jr.

**Methodology:** Lagu A. Androga, LaTonya J. Hickson, Bjoerg Thorsteinsdottir, Sandhya Manohar, Jason K. Viehman, Ziad Zoghby, Suzanne M. Norby, Andrea G. Kattah, Robert C. Albright, Jr.

**Project administration:** Lagu A. Androga, Rachel H. Amundson, Robert C. Albright, Jr.

**Resources:** Lagu A. Androga, Ziad Zoghby, Andrea G. Kattah, Robert C. Albright, Jr.

**Software:** Lagu A. Androga, Jason K. Viehman.

**Supervision:** Lagu A. Androga, LaTonya J. Hickson, Bjoerg Thorsteinsdottir, Ziad Zoghby, Suzanne M. Norby, Andrea G. Kattah, Robert C. Albright, Jr.

**Validation:** Lagu A. Androga, LaTonya J. Hickson, Bjoerg Thorsteinsdottir, Sandhya Manohar, Ziad Zoghby, Robert C. Albright, Jr.

**Visualization:** Lagu A. Androga, Robert C. Albright, Jr.

**Writing – original draft:** Lagu A. Androga, Robert C. Albright, Jr.

**Writing – review & editing:** Lagu A. Androga, Rachel H. Amundson, LaTonya J. Hickson, Bjoerg Thorsteinsdottir, Vesna D. Garovic, Sandhya Manohar, Jason K. Viehman, Ziad Zoghby, Suzanne M. Norby, Andrea G. Kattah, Robert C. Albright, Jr.

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
