## [Decision Letter · Decision Letter 0]

27 Oct 2021

PONE-D-21-23986Telenephrology versus Face-to-face visits: A Comprehensive Outpatient Nephrology Patient Perspective-based Cohort StudyPLOS ONE

Dear Dr. Albright,

Thank you for submitting your manuscript to PLOS ONE. After careful consideration, we feel that it has merit but does not fully meet PLOS ONE’s publication criteria as it currently stands. Therefore, we invite you to submit a revised version of the manuscript that addresses the points raised during the review process. Please address raised comments by both reviewers.

We look forward to receiving your revised manuscript.

Kind regards,

Abhishek Makkar, M.D.

Academic Editor

PLOS ONE

2. Thank you for stating the following in the Acknowledgments/ Funding Section of your manuscript:

“Mayo Clinic, Nephrology & Hypertension Research Committee - $5,000 funding for statistical help”

“Mayo Clinic, Nephrology & Hypertension Research Committee - $5,000 funding for statistical help”

4. Please ensure that you refer to Figure 3 and 4 in your text as, if accepted, production will need this reference to link the reader to the figure.

Reviewers' comments:

Reviewer's Responses to Questions

**Comments to the Author**

1. Is the manuscript technically sound, and do the data support the conclusions?

Reviewer #1: Yes

Reviewer #2: Yes

2. Has the statistical analysis been performed appropriately and rigorously? 

Reviewer #1: Yes

Reviewer #2: Yes

3. Have the authors made all data underlying the findings in their manuscript fully available?

Reviewer #1: Yes

Reviewer #2: Yes

4. Is the manuscript presented in an intelligible fashion and written in standard English?

Reviewer #1: Yes

Reviewer #2: Yes

5. Review Comments to the Author

Reviewer #1: With the current pandemic the need for more telemedicine to serve the population has become more obvious. There are quite a few biases in the results but the authors do a good job of acknowledging them in their discussion whihc includes small survery size, mostly established patients.

Also given that the authors don't know if the survey pertained to specifically the nephrology visit would it not be prudent to change the title to NOT include nephrology?

Also most telehealth visit patients lived further away and are generally known to be more satisfied if they don't have to travel far to see their physician, I think that is worth a mention.

And I like that they acknowledged that they couldn't measure patient outcomes which would be good to look at in the future and see if they correlated with the patient satisfaction scores

Reviewer #2: This is a retrospective single center study using a standardized structured survey to assess patient satisfaction, comparing telenephrology versus face-to-face visits in outpatient nephrology and hypertension clinic. Patient satisfaction was equally high for the telenphrology and face-to-face visits. Telenephrology was more likely to be done for established patients and younger patients. The majority of patients receiving telenephrology were found to reside further from clinic.

These findings were not surprising, but this study confirmed what we experience. There were 52 providers in this study which provided good sampling of the variability among providers. The authors carefully designed the study to assess the survey. The authors also acknowledged the limitations of the study and discussed the future studies that are needed.

Major points

Who decided the modality for each visit, the provider or the patient?

Why did the authors reduce a 5-point Likert scale to a binary scale (i.e., good (4) or very good (5) vs. 1-3)?

It may worthwhile to analyze the repeat visits.

Could the authors further discuss the changes in telenephrology from pre-COVID to COVID times, in addition to the race differences mentioned? Are there any relevant studies that should be cited in this regard?

Minor

Table 2 legend should be October, 2019 - September, 2020.

6. PLOS authors have the option to publish the peer review history of their article (what does this mean?). If published, this will include your full peer review and any attached files.

Reviewer #1: No

Reviewer #2: No

---

## [Author Response · Author response to Decision Letter 0]

17 Feb 2022

Response to Reviewers

Please see below for responses to the reviewers’ comments/ questions

Comments to the Author

1. Is the manuscript technically sound, and do the data support the conclusions?

Reviewer #1: Yes

Reviewer #2: Yes

2. Has the statistical analysis been performed appropriately and rigorously? 

Reviewer #1: Yes

Reviewer #2: Yes

3. Have the authors made all data underlying the findings in their manuscript fully available?

Reviewer #1: Yes

Reviewer #2: Yes

4. Is the manuscript presented in an intelligible fashion and written in standard English?

Reviewer #1: Yes

Reviewer #2: Yes

5. Review Comments to the Author

Reviewer #1: With the current pandemic the need for more telemedicine to serve the population has become more obvious. There are quite a few biases in the results but the authors do a good job of acknowledging them in their discussion which includes small survey size, mostly established patients.

Also given that the authors don't know if the survey pertained to specifically the nephrology visit would it not be prudent to change the title to NOT include nephrology?

Response: All the patients included in our study have kidney disease and visited with our Nephrology & Hypertension division at least once during the study period. These patients commonly have multiple comorbidities requiring care by a multidisciplinary team spanning various medical specialities. As such, we have changed the title of the paper to be “Telehealth versus Face-to-face visits: A Comprehensive Outpatient Patient Perspective-based Cohort Study of Patients with Kidney Disease”

Also most telehealth visit patients lived further away and are generally known to be more satisfied if they don't have to travel far to see their physician, I think that is worth a mention.

Response: This is true. We have now included it in the manuscript (fifth paragraph of discussion section).

And I like that they acknowledged that they couldn't measure patient outcomes which would be good to look at in the future and see if they correlated with the patient satisfaction scores

Response: Understanding patient outcomes, patient and provider perspectives on the different models of telehealth will be very important in figuring out how to improve care delivery to patients. It will be useful to study these factors not just within our institution but nationally and globally too.

Reviewer #2: This is a retrospective single center study using a standardized structured survey to assess patient satisfaction, comparing telenephrology versus face-to-face visits in outpatient nephrology and hypertension clinic. Patient satisfaction was equally high for the telenephrology and face-to-face visits. Telenephrology was more likely to be done for established patients and younger patients. The majority of patients receiving telenephrology were found to reside further from clinic.

These findings were not surprising, but this study confirmed what we experience. There were 52 providers in this study which provided good sampling of the variability among providers. The authors carefully designed the study to assess the survey. The authors also acknowledged the limitations of the study and discussed the future studies that are needed.

Major points

Who decided the modality for each visit, the provider or the patient?

Response: This was mostly based on patient preference. Providers encouraged initial consults to be face-to-face or video. Established patients could choose face-to-face, video or telephone, based on their preference and medical needs. Providers could triage the cases prior to the appointments being set and provide guidance on the appropriateness of the modality, after patients make their preferences known.

Why did the authors reduce a 5-point Likert scale to a binary scale (i.e., good (4) or very good (5) vs. 1-3)?

Response: Our institution uses Likert scale survey questionnaires to evaluate patient experiences. The results are then summarized using top-box score analyses that combines the top two responses to create a single number. The institution focuses on service, quality, and excellence and holds providers accountable to achieving top box score experience when it comes to patient satisfaction.

It may worthwhile to analyze the repeat visits.

Response: At the moment, we do not have enough data, given the study period, to effectively analyze repeat visits. The COVID-19 pandemic also continues to rage on, and this may affect patients’ perspectives on telehealth. However, as telehealth persists and patients continue to seek care at our institution, we will have the longitudinal data and we are excited to share our findings over time.

Could the authors further discuss the changes in telenephrology from pre-COVID to COVID times, in addition to the race differences mentioned? Are there any relevant studies that should be cited in this regard?

Response: Prior to the COVID-19 pandemic, CMS payment requirements limited telehealth reimbursement. As such, many patients and providers did not have access to telehealth. However, COVID-19 pandemic led to the presidential order that directed CMS to expand telehealth services under the 1135 waiver authority and Coronavirus Preparedness and Response Supplemental Appropriations Act. The opportunities to effectively study racial differences in the use of telehealth given access barriers, literacy/language issues along with acceptance is a goal this research team hopes to address in upcoming work.

Minor

Table 2 legend should be October 2019 - September, 2020. 

Response: The study period was from March, 2020 – July, 2020

6. PLOS authors have the option to publish the peer review history of their article (what does this mean?). If published, this will include your full peer review and any attached files.

If you choose “no”, your identity will remain anonymous, but your review may still be made public.

Do you want your identity to be public for this peer review? For information about this choice, including consent withdrawal, please see our Privacy Policy.

Reviewer #1: No

Reviewer #2: No

---

## [Editor Report · Decision Letter 1]

23 Feb 2022

Telehealth versus Face-to-face visits: A Comprehensive Outpatient Perspective-based Cohort Study of Patients with Kidney Disease

PONE-D-21-23986R1

Dear Dr. Albright,

We’re pleased to inform you that your manuscript has been judged scientifically suitable for publication and will be formally accepted for publication once it meets all outstanding technical requirements.

Kind regards,

Abhishek Makkar, M.D.

Academic Editor

PLOS ONE
---

## [Editor Report · Acceptance letter]

1 Mar 2022

PONE-D-21-23986R1 

Telehealth versus Face-to-face visits: A Comprehensive Outpatient Perspective-based Cohort Study of Patients with Kidney Disease 

Dear Dr. Albright Jr:

I'm pleased to inform you that your manuscript has been deemed suitable for publication in PLOS ONE. Congratulations! Your manuscript is now with our production department. 

Kind regards, 

on behalf of

Dr. Abhishek Makkar 

Academic Editor

PLOS ONE